# Assessment and Intervention for Tool-Use in Learning Powered Mobility Intervention: A Focus on Tyro Learners

Lisbeth Nilsson [1,*] and Lisa Kenyon [2]

1    Department of Health Sciences, Lund University, Box 157, 221 00 Lund, Sweden
2    Department of Physical Therapy and Athletic Training, Grand Valley State University,
     301 Michigan Street NE, Grand Rapids, MI 49503, USA; kenyonli@gvsu.edu
*    Correspondence: lisbeth.nilsson@med.lu.se or lisbethsweden@gmail.com

**Abstract:** Young infants, children, and persons of any age who have cognitive limitations can be thought of as tyro learners, who are beginners in learning. For tyro learners, the self-produced mobility afforded by a powered mobility device offers opportunities, to explore tool-use learning and interact with the surrounding environment, thereby potentially enhancing development and learning, providing a foundation for future goal-directed, tool-use activities. The Assessment of Learning Powered mobility use tool, version 2.0 (ALP), developed with tyro learners, is a process-based implementation package, focused on assessing and progressing an individual's understanding of how to use a powered mobility device. Although the ALP tool can be used with any powered mobility learner, research suggests that this process-based approach may be, especially, beneficial for tyro learners, who are in the early phases of learning how to operate a powered mobility device. This article aims to (1) explain tool-use learning in powered mobility intervention; (2) distinguish between the characteristics of process-based and task-based implementation packages; (3) provide an in-depth description of using the ALP tool in providing powered mobility intervention; and (4) highlight the benefits of using the ALP tool, with a focus on tyro learners.

**Keywords:** powered wheelchair; intellectual disabilities; occupational performance; novice; early intervention; education; rehabilitation

## 1. Introduction

Human activity involves many forms of tool use and related tool-use learning [1,2]. The term "tool" can be used in a broad sense to encompass a wide variety of both concrete (hand, switch, joystick, etc.) and abstract (methods for assessment and intervention, etc.) tools [3–6]. Those who are beginners in tool-use learning, young infants, children, and persons of any age who have cognitive limitations, can be thought of as tyro learners (tyro as defined in the Merriam-Webster Dictionary and Cambridge Dictionary). Tyro learners grow consciousness of new activities, starting from the very beginning of the learning continuum [7,8]. For children and adults who are tyro learners, exploring how to use a powered mobility device, such as a powered wheelchair [9–13], modified ride-on toy car [14,15], and other powered devices [16–20], is often less about learning to drive [21,22], and more about Driving to Learn [8,23–25]. The self-produced mobility afforded by a powered mobility device offers opportunities to explore tool-use learning and interact with the surrounding environment [26,27], thereby potentially enhancing development and learning, providing a foundation for future goal-directed, tool-use activities [6–8]. The potential for these benefits supports powered mobility device use in tyro learners, as an effective and appropriate intervention, even though they may never become independent, competent powered wheelchair users [8,28–34].

All individuals, who are learning to use a powered mobility device, pass through a floating continuum, represented by exploratory, operational, and functional powered

mobility learner groups [35]. Tyro learners, often, remain in the exploratory- or operational-learner groups, for prolonged periods of time. As compared to functional-learner groups, exploratory- and operational-learner groups are more dependent on others, to set up and provide powered mobility learning opportunities, and, often, require additional encouragement and motivation, when exploring and learning how to use a powered mobility device [6–8,33–37].

Developed through work with tyro learners [8,25,28,29], the Assessment of Learning Powered mobility use tool, version 2.0 (ALP), is a powered mobility implementation package focused on assessing and progressing an individual's understanding of how to use a powered mobility device [6,7,37]. The ALP tool is comprised of the ALP instrument for assessment and the ALP facilitating strategies for intervention. The ALP instrument defines eight phases in the process of learning (from *Phase 1—Novice to Phase 8—Expert*), encompassed within three stages of learning (*Explore Functions, Explore Sequencing, Explore Performance*). The ALP instrument is used in tandem with the ALP facilitating strategies, such that a learner's actual stage and phase of learning, as identified through the ALP instrument, are used to guide and select the specific ALP-facilitating strategies that will best support the learner. In this manner, the ALP tool provides each learner with the precise level of challenge needed to progress their understanding across the continuum of learning to use a powered mobility device [6,7]. The ALP instrument has been shown to have excellent inter-rater reliability (linear-weighted kappa of 0.85) [38].

As outlined in Table 1, research suggests that the process-based approach [8,28,29], reflected in the ALP tool [7], may be beneficial in supporting powered mobility device use learning in tyro learners, who belongs in exploratory and operational powered mobility learner groups [6,7,35,37]. Task-based approaches [39–46], those involving production of specific powered mobility tasks, may be beneficial for integrative learners focused on integrating both patterns of operation and powered mobility device use into their everyday lives, who are functional powered mobility learners/users [35].

**Table 1.** Comparison of tyro and integrative learners of powered mobility use.

| | Tyro Learners [1] | | Integrative Learners [2] |
|---|---|---|---|
| **Powered Mobility Learner Group** [35] | Exploratory | Operational | Functional |
| **Assessment of Learning Powered mobility use-(ALP) Stage** [7] | ALP Stage 1: Explore Functions | ALP Stage 2: Explore Sequencing | ALP Stage 3: Explore Performance |
| **Recommended Type of Powered Mobility Implementation** [35] | Process-based packages focused on learning and developing a conscious understanding and awareness of how a powered mobility device works | | Task-based packages focused on the production of specific powered mobility tasks, often as listed in a specific protocol |

[1] Young infants and children or persons who have cognitive limitations. [2] Those focused on integrating patterns of operation and powered mobility device use, into everyday life.

This feature article aims to (1) explain tool-use learning in powered mobility intervention; (2) distinguish between the characteristics of process-based and task-based powered mobility implementation packages; (3) provide an in-depth description of using the ALP tool, in providing powered mobility intervention; and (4) highlight the benefits of using the ALP tool with tyro learners.

## 2. Powered Mobility: A Tool-Use Learning Experience

The perception–action experiences provided through engagement in exploration of self-produced mobility are powerful drivers of cognitive developmental change and learning about the world [2,3,13,33,34,47,48]. Numerous researchers have studied powered mobility, as a means for independent mobility [49–54]. For tyro learners, learning by driving a powered mobility device, also, provides the opportunity to explore tool-use learning [6,8,37]. A tool can be thought of as something used to carry out a particular action

or function. Depending on how they are used, a part of our body, an object, and a device can all be tools. Tools range in complexity from simple, one-part tools, such as a spoon, to complex, multi-part tools, such as a smartphone [6,8]. Regardless of what tool is being used, the phases and stages in the process of learning to use a tool are similar in nature [6,7] and involve a growing consciousness, or understanding, of how to use and relate to the tool, when integrating it in everyday activities [7,8].

A powered mobility device is a complex, multi-part tool, involving coordination of three distinct tools: a part of the body (a hand, foot, eye, etc.), an access method (switch/es, joystick, eye-tracking technology, etc.), and the powered mobility device itself [55]. Development of conscious awareness, in how to integrate the use of these three tools together, is a process encompassing learning, as described in the ALP instrument [7]. Controlled execution of actions, with a part of the body (tool 1), and an understanding of the function/s of the access method (tool 2) and the device (tool 3), are necessary to move and control the operation of the powered mobility device (ALP stage 1, explore function). Accurate action on the access method is required to develop a pattern for how to move and control the operation of the powered mobility device in space and time (ALP stage 2, explore sequencing). Functional navigation and integrated use of the powered device in the physical and social environments requires understanding of situations and judgement (ALP stage 3, exploring performance). The path that each learner follows through this tool-use learning process may differ in terms of individual abilities, pace, oscillation across ALP phases, trajectory over time, and motives for development and learning [7,8].

Tyro learners, who are provided with repeated opportunities to explore powered mobility use, may develop their abilities and grow their understanding of tool use [7,8,25,37]. Many tyro learners are motivated to explore, by the experience of independently changing their own position in space [8]. This self-produced change of position provides a new perspective of the situation and changes the tyro learner's relationships to others and objects in their surrounding environment [6,8]. These repeated powerful experiences develop the tyro learner's sense of agency and control, and provide opportunities for acquiring comprehensive knowledge about the world [8,56]. The achievements gained through tool-use learning in a powered mobility device may transfer to other tool-use situations, expanding the tyro learner's opportunities to explore and use other assistive technologies for goal-directed tool-use activity and communication [8,37].

## 3. Distinguishing between Process-Based and Task-Based Powered Mobility Implementation Packages

There are two main types of powered mobility implementation packages: task-based and process-based [7,35]. Both often marry the assessment and intervention procedures, such that assessment directly guides intervention. However, understanding the characteristics of each type of implementation package may assist clinicians/educators in matching the type of package with a specific learners' needs, abilities, and level of tool-use understanding. Table 2 provides an overview of the distinguishing characteristics of the process-based and task-based implementation packages.

### 3.1. Task-Based Powered Mobility Implementation Packages

*Tasked-based* packages, such as the Wheelchair Skills Program (WSP) [45,46], are focused on an individual's production of specific powered mobility tasks (moving the powered mobility device forward, turning the powered mobility device, maneuvering the powered mobility device through a doorway, etc.). In the WSP, the Wheelchair Skills Test for Powered Wheelchairs Operated by Their Users, a standardized assessment instrument, is used to identify specific powered mobility tasks that could be improved with intervention. The accompanying WSP intervention curriculum, known as the Wheelchair Skills Training Program, is used to help learners work towards safe and proficient performance of the identified powered mobility tasks in need of improvement. This task-based approach has been found to be highly successful in adult populations [45,46]. Given its emphasis

on the production of specific tasks, the WSP presumes a foundational understanding of how a powered mobility device operates. As such, integrative learners, who are focused on integrating both patterns of operation and powered mobility device use into their everyday lives, are most likely to benefit from task-based packages [35]. The most prominent characteristics of task-based assessment and intervention are outlined in Table 3.

**Table 2.** Distinguishing process-based and task-based implementation packages.

| | Process-Based Packages | Task-Based Packages |
|---|---|---|
| **Population** | Any learner group, including tyro learners | Operational or functional learner groups |
| **Learner prerequisites** | Awake and consent to involvement (in whatever way they are able to express their agreement) | Aware of how to use access method |
| **Procedure of assessment** | Protocol free<br>Observation, functional analysis, and interpretation of tool-use performance within descriptive categories | Protocol-like<br>Observation of task performance is compared to a description of the task in a protocol |
| **Documentation** | Actual stage and phase of learning<br>Need of facilitating strategies | Pass, fail, or needs improvement<br>Task/s to practice |
| **Intervention goal** | To explore tool-use functions, sequencing or performance | To improve performance of specified tool-use tasks |
| **Intervention setting** | Relevant for the learner's phase of understanding, age, and motives | Relevant for performing the specified tasks |
| **Integration of assessment and intervention** | Concurrent integration<br>Simultaneous assessment and intervention<br>Emphasizes recognition of individual needs and motivation for engagement in the tool-use activity | Linear integration<br>Assessment proceeds intervention<br>Emphasizes tasks specific for the tool being used |
| **Expectations of outcome** | Growing consciousness/understanding of tool use | Safe and controlled tool use |

**Table 3.** Characteristics of task-based implementation packages.

| | Task-Based Assessment | Task-Based Intervention |
|---|---|---|
| **Approach** | Assesses tasks using a protocol-like list of specific tasks to be assessed | Guided execution of tasks |
| **Purpose** | To identify tasks in need of improvement | To develop or refine execution of relevant tasks through practice |
| **Focus of learner's tool-use performance** | Performing tasks specific for skilled tool-use | Practicing to master tasks listed in the protocol |
| **Clinician/educator performance** | Direct observation of execution of task | Instruction<br>Demonstration<br>Repetition of tasks |
| **Focus and relevant knowledge for implementation of package (clinician/educator performance)** | *Assessing WHAT* (task)<br>Following the protocol<br>Descriptive<br>Comparative<br>Concrete, observable actions<br>Pass, fail, or needs improvement | Cognition to contrast performance based on a protocol<br>Complex intellectual processing<br>*Reflection ON action*<br>Giving notice of the obvious<br>Serially adjusting assessment and intervention |

### 3.2. Process-Based Powered Mobility Implementation Packages

*Process-based* packages, such as the ALP tool, focus on assessing and progressing a person's growing consciousness and understanding of how to coordinate and use the combination of three tools (part of body, access method, and device) involved in using a powered mobility device [6,7,55]. In this manner, the ALP tool allows clinicians/educators to apply their knowledge of the learning process, to guide and promote each learner's tool-use understanding, along the broad continuum of learning, to use a powered mobility device, from novice to expert. Although the ALP tool can be used with any powered mobility learner, research suggests that this process-based approach may be, especially, beneficial for tyro learners, who are in the early phases of learning how to operate a powered mobility device [35].

One of the unique characteristics of process-based implementation packages is the emphasis on allowing, through actively supporting, each learner's own initiative within the learning process [7,8,28,29]. Clinicians/educators, thus, must provide the learner with the freedom to explore each of the three tools encompassed within the powered mobility device (part of the body, access method, and device), in whatever ways they desire [6–8,32,55]. Allowing each learner to explore and learn in their own way, and at their own pace, independent of any outside influence or control, provides the learner with the opportunity to figure out for themselves how each of the three tools work, both as separate entities and as a coordinated unit [55]. This, also, enables the learner to experiment with a variety of action patterns and sequences (e.g., using their mouth to activate the joystick, moving the joystick in different directions, experiencing the resulting movement of the device relative to the physical and social environment), and develop their own strategies, while, simultaneously, learning from their own mistakes. Learners are, also, encouraged to develop and act upon their own desires and goals, as they move from exploring the functions of the three tools encompassed within a powered mobility device, to exploring sequencing of these tools and the environment as well as to exploring performance and use of the device to interact with others [7,8,36–38]. This emphasis, on allowing and encouraging the learner to explore and discover in their own way, may help to contribute to a tyro learner's developing sense of self and agency as well as growing autonomy. The most prominent characteristics of process-based assessment and intervention are outlined in Table 4.

**Table 4.** Characteristics of process-based implementation packages.

| | Process-Based Assessment | Process-Based Intervention |
|---|---|---|
| **Approach** | Assesses process of learning and provides indicators, for each stage and phase along the continuum of the tool-use learning process | Tool-use learning is guided by facilitating strategies, for each phase in the tool-use learning process |
| **Purpose** | To identify, in the moment, the learner's actual stage and phase of understanding of how to use the tool | To support the learner's needs and desires based on knowledge of their tool-use understanding, in the specific moment |
| **Focus of learner's tool-use performance** | Allowing learners to explore tool-use in their own way | Practicing at the learner's level of tool-use understanding |
| **Clinician/educator performance** | Continuous functional analysis of learner performance of tool-use | Continuous adjustment, in the moment, of the degree of challenge, to match the learner's actual phase in the learning process |
| **Focus and relevant knowledge for implementation of package (clinician/educator performance)** | *Assessing HOW* (process)<br>Understanding the process<br>Analytic<br>Reflective, relative to described indicators of performance<br>Abstract<br>Continuous assessment during a session | Metacognition to infer learner's tool-use understanding, based on observable actions<br>Higher level of complexity in intellectual processing<br>*Reflection IN action*<br>Discerning the obscure<br>Concurrently aligning the facilitating strategies to actual understanding |

## 4. An In-Depth Description of the ALP Tool

The development of the ALP tool was published by the open-access Journal of Rehabilitation Research and Development (JRRD), and the complete ALP tool is provided as an online resource to Nilsson and Durkin on their webpage [7]. The ALP instrument is found in the JRRD online Appendix 1, and the ALP facilitating strategies are found in the JRRD online Appendix 2 [7]. A brief overview of the prominent elements, of the ALP instrument and ALP facilitating strategies, for each of the three ALP stages, are outlined in Table 5 (ALP stage 1), Table 6 (ALP stage 2), and Table 7 (ALP stage 3).

**Table 5.** ALP tool: prominent elements of stage 1: explore functions—phases 1–3 *.

| ALP Phase | Attention | Activity and Movement | Understanding of Tool-Use | Expressions and Emotions | Interaction and Communication |
|---|---|---|---|---|---|
| **ALP Instrument, Stage 1: Explore Functions: Body and Tool(s)** | | | | | |
| **1** **Novice** | Extreme distractibility | Excited, Non-Act, Rejection | No idea or vague idea of tool use | Open, Neutral, Anxiety | No response, Avoidance |
| **2** | Single channeled At times more alert, at times, passive | Pre-act | The idea of basic use is born | Contented, Curious, Anxious, Angry | Responds to interactions |
| **3** | Single channeled attention but able to shift attention | Acts directed | Basic use of the tool | Serious, Contented, Smiles | Initiates interactions |

**ALP Facilitating Strategies, Stage 1—Introvert Level**

*Use a gentle approach to establish a safe and secure relationship*
*Focus attention to tool function and close vicinity*

**Oriented at tool use functions and tool interaction**

- Persistence with interactive approaches and physical demands, such as touch and manual guidance
- Build awareness of body use for tool use; encourage exploration of tool parts and tool function
- Offer manual guidance, adjusting according to learner's needs
- Constantly shift between withdrawing and re-entering the learner's space

**Frustration**

- Accept the learner's rejection, anxiety, or passivity and acknowledge there may be signs of basic frustration at this stage

**Oriented at social interaction**

- One-to-one interaction
- Structured variation of physical and social interaction

**Language**

- Imperative and deliberate, but gentle, slowly paced language
- Labeling of body parts, tool parts, acts, and effects
- Attaching words to the acts being performed
- Precise and condensed language

**Encourage own initiatives to act**

- Inspire own initiatives by using manual guidance
- Allow the learner to do trials their own way
- Allow repetitions of exploratory acts to the learner's level of satisfaction

* Full ALP tool, version 2.0, see Nilsson and Durkin [7], online Appendix 1 and 2, at JRRD's homepage.

Foundational concepts within the ALP tool relate to promoting an atmosphere of mutual respect and collaboration with the goal of empowering the learner [7,8,36–38]. The clinician/educator sets up as well as prepares the environmental and social circumstances for powered mobility assessment and intervention, to meet the specific needs of the learner and invites the learner to engage in an allowing co-constructive partnership [6].

As noted in Tables 5–7, the ALP instrument for assessment defines eight phases of learning, from Phase 1 (Novice) to Phase 8 (Expert). Categories of observation with each ALP phase include Attention, Activity and Movement, Understanding of Tool Use, Expressions and Emotions, and Interaction and Communication. These five categories, with their indicators, structure the clinician's/educator's observations of the learner's occupational performance (behaviors, acts, actions, emotional expressions, interactions, communication).

In this manner, the clinician/educator is able to infer the learner's actual phase of tool-use understanding [6,7,38,57]. The "Attention" category reflects the learner's ability to regulate and shift their attention. "Activity and Movement" focuses on the learner's performance of activity and motor control. "Understanding of tool use" pertains to the learner's level of consciousness/understanding, regarding the operation of the powered mobility tool/s and is the primary, observable cognitive component, within each ALP phase. "Expression and Emotions" provides insights into the learner's motivation, whilst "Interaction and Communication" addresses the expression of interpersonal relationships and social interplay amongst the powered mobility device, the environment, and available social partners. Indicators for each of these observational categories are described in detail, for each of the eight ALP phases, in JRRD online Appendix 1 [7].

**Table 6.** ALP tool: prominent elements of stage 2: explore sequencing—phases 4–5 *.

| ALP Instrument Stage 2: Explore Sequencing: Body, Tool(s), Environment | | | | | |
|---|---|---|---|---|---|
| ALP Phase | Attention | Activity and Movement | Understanding of Tool-Use | Expressions and Emotions | Interaction and Communication |
| 4 | Single-channeled attention but able to shift spontaneously | Chains of acts | Exploration of extended tool use | Serious, Smile, Sometimes laughs | Mutual interactions |
| 5 | Two-channeled attention | Sequences of chains of acts | Idea of competent tool-use is born | Eager, Smiles, Serious, Frustration | Reciprocated interactions, Triadic Interaction |

| ALP Facilitating Strategies, Stage 2—Difficult Transition |
|---|
| *Focus attention to tool function and external tool use goals* |

**Oriented at tool use interaction**
- Encourage exploration and experimentation with pattern building
- Non-interference intertwined with manual guidance and verbal prompts or instructions
- Introduce speed variation and the concept of using gentle, slow, graded body movements with the operating tool–**joystick, switch or other access method**
- Confirm success, difficulties, and failure of patterning details
- Provide external motivators, to convey the idea of a goal for tool use

**Frustration**
- Accept the learner's expressions of frustration
- Calm and reassure the learner, to reduce their level of frustration

**Oriented at social interaction**
- Extend interplay distance
- Introduce simple playful one-to-one interactive activities and, then, gradually increase the complexity

**Language use**
- Dialogue using simple verbal language, body language, and guiding
- Labeling and explaining tool function and tool-use outcome

**Encourage own initiatives and trials—find out how it works, give it another try, try another way!**
- Allow the learner to explore use of the tool, in their own way
- Allow the learner to make their own mistakes
- Allow the learner to develop their own strategies

* Full ALP tool, version 2.0, see Nilsson and Durkin [7], online Appendix 1 and 2, at JRRD's homepage.

As illustrated in Tables 5–7, the ALP phases are encompassed within three stages of learning: stage 1: explore functions (ALP Phases 1–3); stage 2: explore sequencing (ALP Phases 4–5); and stage 3: explore performance (ALP Phases 6–8). Learners in ALP Stage 1 (Table 5) are in an introverted stage of learning and may appear shy and withdrawn. As such, they need time and understanding, to be able to explore the functions of their body in relation to the access method and the powered mobility device as well as to learn concepts related to cause and effect Learners in ALP Stage 2 (Table 6) are in the difficult transitional stage, of exploring the sequencing of the inter-related functions of their body, the powered mobility tools, and the environment. They may appear eager, but will often exhibit frustration because their understanding of what they can do with the powered mobility device outpaces their ability to achieve what they desire to accomplish.

Learners in ALP Stage 3 (Table 7) are in the extrovert stage of learning and exploring performance, so are learning to integrate powered mobility use into their daily activities. These learners are developing and demonstrating higher cognitive functions, within goal-directed powered mobility activities, specific to the learner's actual ALP phase, within this stage. A flowchart outlining use of the ALP tool is provided in Figure 1, below.

**Table 7.** ALP tool: prominent elements of stage 3: explore performance—phases 6–8 *.

| ALP Phase | Attention | Activity and Movement | Understanding of Tool-Use | Expressions and Emotions | Interaction and Communication |
|---|---|---|---|---|---|
| **ALP Instrument, Stage 3: Explore Performance: Body, Tool(s), Environment, Occupation** | | | | | |
| **6** | Multi-channeled attention but easily disrupted | Goal-directed activity | Competent use of the tool | Serious, Content, Laugh, Exited | Consecutive interactions |
| **7** | Multi-channeled attention, generally focused | Occupation, for its own sake | Fluent, precise use of the tool | Happiness, Satisfaction | Concurrent interactions |
| **8 Expert** | Attention well established and sustained | Occupation composed of two or more activities | Integrated tool use | Dependent on the doing of "other" activities | Multi-level integrated interactions |

**ALP Facilitating Strategies, Stage 3—Extrovert level**

*Focus attention to competent tool use and physical and social environment*

**Oriented at tool use interactions**

- Encourage exploration of tool use, in everyday environment

**Frustration**

- Facilitate the learner's development of strategies, to release blocking and decrease frustration

**Oriented at social interaction**

- Unstructured variation
- Group interaction

**Language use**

- Verbal dialogue; labeling, reasoning, instructions for exercises, tasks
- Mutual agreements, facilitate judgement skills, care of self and others

**Promote the learner's own initiatives**

- Encourage development of own desires, goals, and initiatives for tool use
- Allow exploration the learner's own way with own mistakes
- Allow unsafe operations, to a certain extent
- Allow the learner to take the lead

* Full ALP tool, version 2.0, see Nilsson and Durkin [7], online Appendix 1 and 2, at JRRD's homepage.

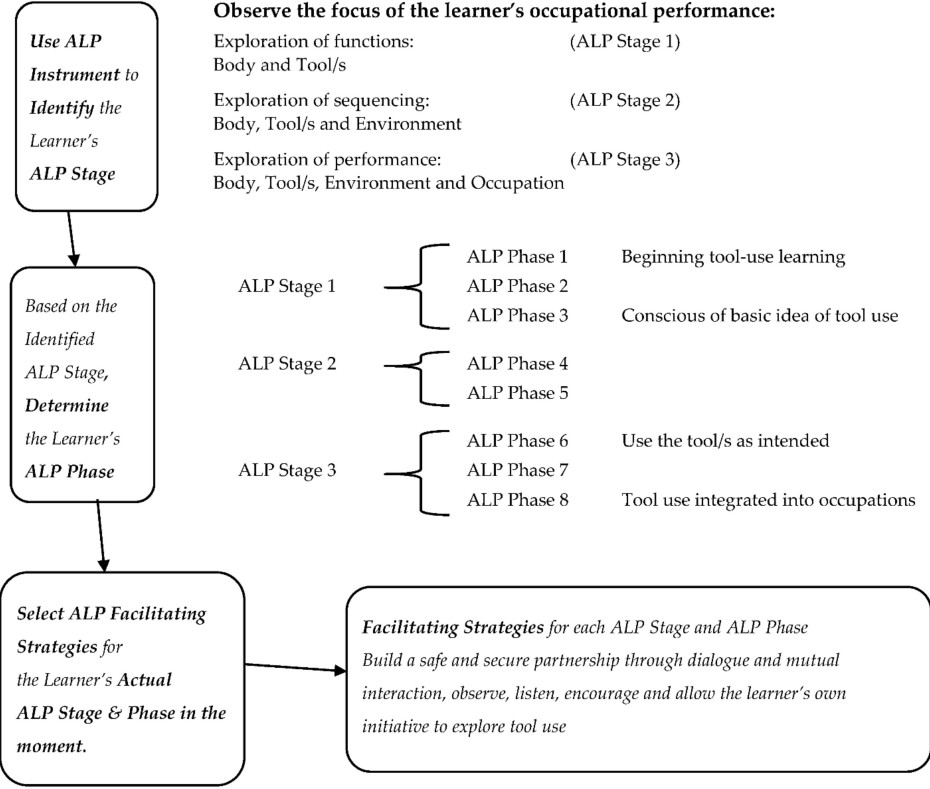

**Figure 1.** Flowchart showing how to use ALP-instrument and ALP-facilitating strategies.

The goal with the first session of powered mobility use is to identify where the learner is in the learning process. During this first session, observations of the learner's performance are interpreted, to recognize and determine the learner's most stable or dominant ALP stage and phase, within the session. When determining a learner's ALP stage and phase, it is recommended that clinicians/educators use a printed copy of the complete three-page ALP instrument [7] (Appendix 1), beginning by identifying the learner's ALP stage.

Then, using a highlighter or other colored pen, the clinician/educator can highlight the observed indicators of performance within, and even across, the specified phase(s). Observations may be scattered across two or more phases, as oscillation in the learner's tool-use learning understanding is, commonly, observed during a single session of powered mobility device use, due to variation in the learner's attention and energy levels. In these instances, the clinician/educator should select the learner's most stable or predominant ALP phase. Once the ALP phase is identified, the clinician/educator consults the ALP-facilitating strategies [7] (Appendix 2), to select specific strategies that will support the learner.

This most stable and dominant ALP stage and phase is used to guide the setup of the intervention situation in the subsequent session [7]. From there on, the learner's actual ALP phase in the precise moment during the intervention session guides how the clinician/educator adjusts the challenges in the situation. It is important for the clinician/educator to stay vigilant to as well as follow the learner's oscillating attention and understanding of the learning situation.

## 5. Using the ALP Tool with Tyro Learners

Although the ALP tool is recommended, when engaging young children in powered mobility intervention [34,36,58], tyro learners of any age may begin their exploration of powered mobility use at earlier phases in the learning process [7]. They may, also, remain in each phase for prolonged periods of time [8,25,37,38]. Supporting tyro learners in the earlier phases of the learning process may require a more individualized intervention approach, as compared to supporting integrative learners who are in the later phases of tool-use learning [7,31,35]. For example, in the early phases, tyro learners need support to explore what the tool can be used for and how it can be handled in a goal-directed way, whereas, in the later phases, integrative learners need support to explore how the tool can be integrated into everyday activity and participation.

Determining a learner's actual stage and phase of learning, to use a powered mobility device, allows clinicians/educators to optimally support and guide the tyro learner's development of tool-use understanding [7,37,38]. As the ALP tool covers the entire learning process, there are not specific prerequisites or requirements that must be met, before a tyro learner engages in the exploration of learning to use a powered mobility device. To initiate powered mobility device learning, the learner must be awake and agree to engage in the activity, as indicated in whatever way they are able to express their agreement. The powered mobility device is, then, introduced to the learner in a manner and pace that supports the learner's specific needs and preferences for approaching new situations. Ensuring that the learner feels safe and secure, both emotionally and physically, is paramount. Once the learner is ready to be lifted into and placed in the powered mobility device, they should be encouraged to explore the device and the access method, in their own way. The learner may need time to adjust to new situations and overcome any uncertainties or anxiety. Simple verbal explanations of what can be done with the powered mobility device as well as short verbal prompting or gentle manual guidance may be used to encourage the learner to explore the device functions, in a way that fits their abilities and motives. The resulting learner performance within this initial session is assessed, using the ALP instrument, to determine the learner's actual consciousness/understanding of tool use. The identified predominant ALP stage and phase are used to select the ALP facilitating strategies, which will guide the powered mobility intervention for the specific learner, as illustrated in Tables 5–7.

*Benefits of the Process-Based ALP Tool*

One benefit of using the ALP tool is that within its eight phases, the ALP tool encompasses the entirety of the learning continuum [7,35,37]. The only prerequisites for beginning powered mobility intervention are that the learner must be awake and agree to engage in the activity, as indicated in whatever way they are able to express their agreement. This open, flexible approach not only allows tyro learners, but also integrative learners, to enter the tool-use learning process, at any point along the learning continuum. As such, there are no limitations of when to begin powered mobility intervention.

Within the ALP tool, the focus for assessment of the learner's tool-use performance is to identify the phase of tool-use understanding and use this information to guide the selection of appropriate facilitating strategies for intervention [7,38]. This focus on process allows clinicians/educators the freedom to implement powered mobility intervention sessions, in ways that align with each learner's actual needs, regardless of their age, abilities, diagnosis, or culture. Since each learner is their own individual, the learner's personal abilities, needs, and particular psycho-social and physical environment determine where and how to carry out the intervention as well as which specific challenges will best meet each learner's characteristics and circumstances. In this manner, the clinician/educator tailors the learning challenges, in accordance with the individual's ever-changing abilities, motives, and actual understanding of powered mobility use, with the end goal of growing consciousness/understanding of tool use [8].

By understanding the phases in the process, through which tool-use learning occurs, clinicians/educators can adjust their intervention approaches to any learner's actual phase of tool-use understanding. There are very few assessments that have been developed with or for tyro learners [59,60], and there is a need for valid and reliable assessments, with a sensitivity to identify small changes in tool-use performance, in children with physical and multiple disabilities [5]. Therefore, knowledge of the learning process provided by the ALP tool is essential for achieving optimal outcomes, especially when engaging tyro learners in intervention involving powered mobility use [6–9,25,32,36,61]. In addition, researchers and practitioners performing powered mobility interventions with both tyro and integrative learners have applied the ALP tool in their work [33,62–66].

## 6. Conclusions

This feature article has highlighted the benefits of engaging tyro learners in exploration of tool-use learning in powered mobility intervention. It distinguishes the characteristics of process-based and task-based powered mobility implementation packages and provides an in-depth description of the process-based ALP tool, to assist clinician/educator in choosing the assessment and intervention approach that best meets the needs of a specific tyro or integrative learner. Research exploring and expanding the application of the ALP tool version 2.0 for powered mobility use, as well as the transferability, relevance, and usefulness of the new generic Assessment of the Learning Process, ALP tool version 3.0 for general tool-use learning (Nilsson unpublished data, [67]) are needed to determine the relevance of these assessment and intervention tools in different learner groups and within various tool-use-learning situations.

**Author Contributions:** L.N. received the invitation and enrolled L.K. in the co-construction of the manuscript. The two authors, in collaboration, developed the structure and organization of the content, elaborated on the presentation of the aims and message, revised and edited the manuscript with tables and figure, and provided final approval of the version to be submitted. All authors have read and agreed to the published version of the manuscript.

**Funding:** This research received no external funding.

**Institutional Review Board Statement:** Not applicable.

**Informed Consent Statement:** Not applicable.

**Conflicts of Interest:** The authors declare no conflict of interest.

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
