# Peer review of "Assessment and Intervention for Tool-Use in Learning Powered Mobility Intervention: A Focus on Tyro Learners"

_disabilities, doi:10.3390/disabilities2020022_

Round 1
Reviewer 1 Report
Frontiers Review: disabilities-1646171
Assessment and intervention for tool-use in learning powered 2 mobility intervention: a focus on tyro learners
Lisbeth Nilsson and Lisa Kenyon
Thank you for the opportunity to review this manuscript. This invited feature paper discusses the assessment and intervention for tool-use in learning powered mobility interventions, with a focus on tyro learners. The authors define tyro learners as young infants, children, and persons of any age who have cognitive limitations who are beginners in learning. They suggest that the experience of self-produced mobility through the use of a powered mobility device promotes opportunities to explore tool-use learning and lays the foundation for future tool-use activities. The manuscript’s aims are to (1) explain tool-use learning in powered mobility intervention; (2) distinguish between the characteristics of process-based and task-based powered mobility implementation packages; (3) provide an in-depth description of using the ALP tool in providing powered mobility intervention; and (4) and highlight the benefits of using the ALP tool with focus on tyro learners.
The authors succeed in accomplishing their aims and in the process, contrast differences between process-based and task-based powered mobility implementation packages for tyro and integrative learners.
Although the manuscript is relatively clear and organized, I was confused by the use of the word ‘tool’ in several different contexts (in both the abstract and throughout the main text). I suggest adopting different words along with clearer explanations to make it easier for the reader to follow the author’s intentions, and then ensure they are used consistently throughout the text. Greater consistency throughout the text would also be of benefit for other terms, most notably ‘power mobility’ vs. ‘powered mobility’.
As for Key Words, there are one too many - the author guidelines state a maximum of 8. I wonder about ‘electric mobility device’ and ‘powered mobility learner groups’ as common searchable terms?
There are a number of areas where revision may improve readers’ understanding and ease of reading:
Line 14 and Line 36 Suggest adding ‘and’ between learning and providing
Lines 30-31 The second part of the first sentence doesn’t make sense (and grow consciousness of new activities starting from the very beginning of the learning continuum). Suggest separating the two concepts presented in that one sentence.
Lines 49-57 and Lines 78-89 Clarification needed that the ALP instrument (assessment form) as well as the ALP facilitation strategies make up the ALP implementation package. The way it is written is very confusing since tool and instrument are often used interchangeably (i.e. referring to the measurement tool). Adding to the confusion, learning tool use (the cognitive process) is one of the main constructs being discussed (first paragraph of Page 3), and this is very different from the measurement ‘tool’. I suggest (i) deleting ALP ‘tool’ and instead use ALP instrument to refer to the assessment form, (ii) use ALP implementation package when discussing both the instrument and facilitation strategies (iii) leave ‘tool use’ as is.
It may be worthwhile to consider introducing/explaining what is meant by an implementation package earlier in the text.
Table 1 Last row - reference 15 does not specifically identify ‘Type of Powered Mobility Implementation Package’ Suggest removing this reference or change the terminology to reflect the content of that article more accurately.
Lines 90-100 In addition to my comment above for lines 49-57 and lines 78-89, there is further confusion when the idea of three tools are introduced (a part of the body, an access method, and the motorized device). Suggest using three ‘components’ instead (the first sentence identifies a powered mobility device as a complex, multi-part tool and having 3 tools as part of a multi-part tool also doesn’t make sense).
Line 99 Delete ‘the’ before powered device
Line 100 Add ‘s’ to environments
Table 2 In third row, under Procedure of assessment, I am unclear as to what is meant by ‘Observation of task performance as compared to task execution as described in a protocol.’ Nor is this is explained further in the text.
Table 3 In last row under Focus and relevant knowledge for implementation of package (clinician/educator performance), delete ‘or’ between pass or fail and add comma after pass (to be similar to Table 2)
I have difficulty understanding what specifically is meant by ‘concrete’, ‘cognition’, and ‘complex’. I also think this is too simplistic an analysis of task-based assessments and interventions, (notably with the statement ‘Consecutively adjusting assessment and intervention’). Further explanation is suggested within the text.
Lines 116- 124 The heading refers to intervention packages whereas the rest of the paragraph references implementation packages-suggest reviewing content to ensure consistency in terminology throughout text.
Lines 147-155 Delete ‘tool’ in reference to ALP on lines 147, 150 and 153 – and follow suggestions stated above.
Line 157 Delete ‘allowing and’ and the comma after supporting
Line 167 Delete ‘own’ (redundant with their)
Line 159-170 Change reference to three tools (as above for Lines 90-100)
Line 172 Delete ‘allowing and’
Table 3 & 4 It might be useful to combine contents of Tables 3 & 4 to make it easier to compare characteristics between process-based and task-based assessment and interventions. Also following comments of consistent terminology, Tables 3 and 4 refer to assessment and interventions whereas earlier references are to instruments and facilitation strategies.
Table 4 First row under intervention: change statement to ‘guided by facilitating strategies for each phase in the tool-use learning process’ to be consistent with other column and Table 3
Under purpose- suggest changing ‘actual’ to current’ as well as replacing ‘in the specific moment’ with ‘current’
Under clinician educator performance- suggest changing ‘actual’ to current’ (also in next row- and in remaining areas in text)
Under Focus and Relevant knowledge… wonder why nothing is italicized but it is in Table 3?
Also wonder what is meant by reflective, relative, and metacognition and higher complexity?
Suggest that like Table 3 there is some oversimplification of concepts presented.
There may also be overlap of some concepts for task-based packages (for example metacognition)
Line 179-180 Suggest changing ‘tool’ after ALP in keeping with comments above
Tables 5,6,and 7 I am curious if permission has been granted from the Journal of Rehabilitation Research and Development (JRRD) to include such a significant chunk of the ALP assessment and facilitation strategies from the original publication? I think the manuscript could stand on its own without these three tables.
Line 227 Exchange ‘tools’ for device
Line 232 Insert ‘the’ prior to powered mobility device
Figure 1 Suggest substituting ‘current for ‘actual’ in bottom left bubble
Suggest changing statement in bottom right bubble to ‘Facilitating Strategies
Empowering Approach for each ALP Stage and Phase:’
Line 272 Suggest using ‘instances’ instead of ‘stances’
Line 274 Delete ‘the’ specific strategies
Sections 5 and 5.1 There seems to be some duplication with this and earlier content- suggest revision to minimize overlap
Lines 284-286 This sentence does not make sense as it is written. Suggest separating for ease of reading. (e.g. It is also useful for tyro learners of any age when beginning their exploration of of tool-use at earlier phases in the learning process. They may too remain in each phase for prolonged periods of time).
Line 294 Delete ‘As previously mentioned’
Line 296 Delete ‘tool’ and substitute implementation package
Line 297 Delete ‘any’
Line 304 Delete ‘placed’ and change ‘into’ to ‘in’
Line 306 Delete ‘initially’
Line 317 Delete ‘tool’
Line 321 Suggest ‘This open, flexible approach not only allows tyro but also integrative learners to enter’
Line 324 Suggest delete ‘tool’ and ‘of tool use performance’
Line 329 Delete ‘Because each individual is their own individual’
Line 339 Insert ‘an’ before optimal outcome
Line 353 Change ‘tools’ to either instruments or implementation packages

Author Response
Please see the attachement

Reviewer 2 Report
It is judged to be a useful content that follows the general form of feature research. As it is a research topic in an area that is not commonly encountered, it seems that the research value is high in terms of scarcity and specificity. Although the expression tyro learner is an expression that is not well used in North America or rehabilitation medicine, there was no shortage in understanding the content. However, as a publication type, the identity located at the intersection of the development paper and the experimental paper or clinical research is sparse, so it seems insufficient to provide clinical reasoning or evidence for what the authors are eventually trying to implement. I'm sorry that the whole story may seem complicated. It may have been wiser to suggest a correlation or direct association between ALP and WST-power wheelchair. It is very important for research to limit the scope of the subject. It may be a conservative approach, but the identity of the study begins with the limitation of the subject range, so the results of this study are regrettable in this respectively. The disadvantage is that there are many ambiguous parts from the standpoint of clinicians who want to apply them to clinical trials, and there are awkward parts because of the conclusion that results in a simple result compared to the vast algorithm. However, if this methodology is valid in hospital settings in Europe and elsewhere, it will be difficult to denigrate. In future studies, it would be better to organize the contents more accurately, strictly, and systematically. Thank you for the paper.
Author Response
Please see the attachement

Reviewer 3 Report
Thank you for the opportunity to review the manuscript.
The introduction of the term- "tyro" is new but does not add to the knowledge base, only replaces the term "exploratory" (by Field and Livingstone in 2018).
The manuscript organizes in one article the stages and facilitating strategies per stage without providing addition knowledge. All of this knowledge has been previously published.
Author Response
Please see the attachement

Round 2
Reviewer 3 Report
Thank you for the opportunity to revise the manuscript.
As per your comment, you added many references (58 to be exact), giving references to statements in a broad way- for example page 1- line 35 and line 26 and page 2 line 73. I think the editors need to decide if this okay.
Page 9- line 212- says " as seen in tables 5 and 6 there are 8 phases" tables 5-6 cover phases 1-5.
Figure 1- repeats text in tables. I don't think you need the figure- or cut from text.
Author Response
Thank you for your feedback on our additions of references. You and the editors has helped out with the revision by pointing out issues that need to be contemplated. You also recognized a misspelling in the text.
We still want to elucidate the perspective of providing powered mobility as a tool-use learning intervention – driving to learn – not only as an intervention to achieve independent mobility. This perspective is of paramount importance for the tyro learners focused in the manuscript.